# Children with ADHD Symptomatology: Does POET Improve Their Daily Routine Management?

**DOI:** 10.3390/children10061083

**Published:** 2023-06-20

**Authors:** Carmit Frisch, Emanuel Tirosh, Sara Rosenblum

**Affiliations:** 1The Laboratory of Complex Human Activity and Participation (CHAP), Department of Occupational Therapy, Faculty of Social Welfare & Health Sciences, University of Haifa, Haifa 3498838, Israel; rosens@research.haifa.ac.il; 2The Hannah Khoushy Child Development Center, Bnai Zion Medical Center, Faculty of Medicine, Technion-Israel Institute of Technology, Haifa 69094, Israel; emi.tirosh@gmail.com

**Keywords:** early intervention, attention deficit hyperactivity disorder, executive function, POET

## Abstract

Children with attention-deficit/hyperactivity disorder (ADHD) struggle with executive delays while managing their daily tasks. This is a secondary analysis of existing data from open-label research examining the efficacy of Parental Occupational Executive Training (POET). It further examines POET’s efficacy in increasing young children’s (3.83 to 7.08 years) executive control over daily routines, and in decreasing their ADHD symptoms. Additionally, the second analysis investigates which of the children’s increased capabilities is better associated with the change in their daily routine management following the intervention. Parents of children with ADHD symptomatology (N = 72, 55 boys) received eight POET sessions. They completed standardised ADHD symptomatology, executive management of daily routines, and executive functions (EF) questionnaires at pretest, post-test, and 3-month follow-up. Children’s ADHD symptoms and their management of daily routines significantly improved following the POET intervention. The children’s score changes in EF accounted for 37% of the variance in their improved routine management. These findings suggest that interventions aiming to increase children’s executive control over their daily routines should improve their broader array of EF besides decreasing core ADHD symptoms.

## 1. Introduction

ADHD is diagnosed based on three core symptoms derived from delayed inhibition: inattention, impulsivity, and hyperactivity [1]. In the past two decades, these symptoms have been considered part of a wider mechanism: executive functions (EF; [2]). EF are mental faculties that are developmentally delayed in children with ADHD and thus may decrease children’s ability to efficiently manage their daily functioning [3]. Because EF are central to gaining self-control over behaviour, functioning, movement, and speech, the literature recommends conducting directing interventions for children with ADHD symptomatology to improve their EF [4,5,6,7] and functional impairments [8]. Children’s increased ability to self-manage their daily routines strongly predicts general and work self-efficacy later in life [9]. Therefore, to enhance the well-being of children and their families, interventions for children with ADHD symptomatology should focus on improving children’s delays in the wide array of EF and improving efficient management of their routine daily tasks. When treating preschool children with ADHD symptomatology, the literature recommends applying the intervention by evidence-based parent training (PT), which has a better impact than direct intervention in this age group [1]. PT’s main advantage is addressing behavioural management and functions in addition to the core ADHD symptoms. Behavioural PTs are also most effective because they help parents adjust and improve their own behaviours and thus increase their ability to manage their children’s oppositional, defiant, and noncompliant behaviours [3].

Few behavioural PTs for children with ADHD symptoms aim at improving the EF of school children. For example, PATHKO is a performance-based intervention that focuses on ameliorating children’s organisation, time management, and planning skills. It was found to be of promising clinical utility in improving daily functioning that requires OTMP skills among children with ADHD [10]. However, to the best of our knowledge, there has been no reported evidence-based PT to conceptually promote EF and their consequent implications on the daily functioning management of pre-schoolers [11]. Forty PTs for children aged 3 to 15 years, examined by Lee, Niew, Yang, Chen, and Lin [12], aimed at changing the children’s behaviour and parents’ behaviours and perceptions. Other studies presented PTs focusing on reducing children’s ADHD symptomatology [13]. All the above PTs did not improve the children’s wide array of EF [14].

POET is an innovative intervention for young children with ADHD symptomatology that aims to promote EF and their consequent implications on daily functioning management. It was developed as an integrated approach that combines performance-based practice with skill acquisition in children’s natural environment. Applying Barkley’s EF terminology [3], POET promotes using cognitive strategies that facilitate children’s cognitive and behavioural inhibition, working memory, self-direction of emotions and motivation, flexible planning, and problem solving in functional contexts. Our primary analysis of POET’s efficacy showed a significant improvement in children’s daily functioning and EF, as well as in their parents’ knowledge regarding EF and tools to cope with delayed EF following the intervention [11]. However, we have not yet examined whether the POET intervention indeed improves children’s ability to manage their daily routines, and whether this improvement is related to children’s improved EF following the intervention.

Therefore, in the current study, we had two primary research questions. First, we examined whether POET resulted in increasing children’s ability to manage their daily routines efficiently, and in reducing their ADHD symptoms. Second, we examined which of the children’s increased capabilities following the intervention was better associated with the observed improvement in managing their daily routines. To this end, we performed a secondary analysis of existing data regarding POET’s efficacy in improving children’s EF, measured using the Behavioural Rating Inventory of Executive Functions/Preschool (BRIEF/P). (1) We hypothesised that following the POET intervention, the study participants would show (a) a significant increase in their ability to manage their daily routine tasks efficiently and (b) a significant reduction in their ADHD symptoms. (2) Then, we hypothesised that the children’s improved EF scores [11] would account for higher percentages of the variation in children’s ability to manage their daily routines following the intervention.

## 2. Materials and Methods

### 2.1. Participants

We calculated the required sample size for a pre–post-intervention design that includes two groups (study and comparison) and four measurements. The calculated sample size was 80 participants using the G Power program with an effect size of 0.30, significance of 0.05, and power of 0.80. A convenient sample was recruited. Between March 2013 and February 2015, public child development centres and private occupational therapy clinics in Israel’s North and Central Districts referred the study families whose concerns about their children’s daily functioning raised suspicion of executive delays. The inclusion criteria were children who met the DSM-5 (American Psychiatric Association (APA), 2013) criteria for ADHD with *t* scores of at least 65 on one or more of the Conners’ Parent Rating Scale (CPRS) and Conners’ Teacher Rating Scales (CTRS) attention or hyperactivity–impulsivity subscales (Goyette, Conners, and Ulrich, 1978), as assessed by the first author after reviewing their questionnaires. We included only children who learned in mainstream classes with no evidence of developmental delays in language understanding or emotional diagnoses according to their parents’ report. Children had not necessarily received the diagnosis of ADHD at the point of inclusion. They did not receive additional treatment during the study period. Children with chronic illnesses or receiving chronic or short-term medication or nonmedical interventions were excluded. To support the parents’ ability to apply the intervention, we included only children from two-parent families and excluded children whose parents reported self-diagnoses of anxiety/depression. Of 116 families referred to this study between April 2013 and February 2015, 72 children (62.1%) were included based on the inclusion and exclusion criteria. Participating children were aged 3.83 to 7.08 years (M = 5.42, SD = 0.86). Due to the low recruitment rate, and as the early results were positive, we ended the study with 72 participants.

### 2.2. Measures

All measures were used in their Hebrew versions and were validated in the current sample. For more details, see [11].

#### 2.2.1. Screening and Background Characteristics Questionnaires

The demographic questionnaire collected sociodemographic and health-related data about the children and their family members, such as age, date of birth, gender, parents’ education, developmental difficulties, and health status. The screening and background characteristics questionnaires are described in more detail in [11].

Following the method applied by [15], an 18-item questionnaire was created based on the DSM-IV text revision (TR) criteria [16]. The questionnaire was used to identify ADHD symptoms and screen the children for inclusion in this study. The DSM-IV-TR criteria for ADHD were translated into Hebrew and adapted for preschool children. Three expert occupational therapists and a senior neurologist established expert validity for this questionnaire version (see [11,17] for more details).

#### 2.2.2. Outcome Measures

CPRS was used to validate the children’s diagnosis of ADHD symptoms by gathering information about specific ADHD subscales [18], and as a measurement tool. The CPRS includes 48 items related to six subscales: conduct problem, learning problem, psychosomatic, impulsive–hyperactive, anxiety, and hyperactivity. Items are ranked on a 4-point scale from 0 (not present) to 3 (often present), and the scores are converted to *t* scores. The CPRS Cronbach alpha values in the current sample were 0.882 for all 48 items, 0.718 for the conduct subscale, 0.644 for the learning subscale, 0.713 for the psychosomatic subscale, 0.673 for the impulsive–hyperactivity subscale, and 0.602 for the ADHD subscale. The Cronbach’s alpha of the anxiety subscale was 0.344.Executive Function and Occupational Routine Scale. The EFORTS is a parental questionnaire, designed to measure children’s ability to manage their daily routines efficiently. It includes 30 items related to three daily routines: morning–evening, play–leisure, and social routine. The EFORTS items concern a child’s ability to apply different EF in the context of performing activities that are usually included in the above three routines. For example, parents rank their child’s ability to persist at an appropriate pace while getting dressed, getting prepared for bed, or playing. They also rank their child’s ability to guide his/her own behaviours during the three routines according to the expected sequences/rules, to solve problems arising in specific activities, or to inhibit strong emotions while playing with a friend. All items are ranked on a Likert scale from 1 (never) to 5 (often). The raw scores are interpreted according to the EFORTS cutoff score for the two age groups (3–5 years, 6–11 years). The EFORTS has high internal reliability for its three factors (ranging from 0.83 to 0.92) and final score (α = 0.947). It has face validity and convergent validity with the BRIEF/P. Its construct validity was derived from exploratory and confirmatory factor analysis with good fit measures: comparative fit index = 0.90 and root mean square error of approximation = 0.06. Finally, the EFORTS has cut scores with means and standard deviations for each daily routine [19], and the reliability obtained in the current sample was acceptable (α = 0.72–0.89).BRIEF/P is a parental questionnaire that measures EF. We used the preschool (BRIEF-P) version for children up to 5 years and 11 months old and the BRIEF for older school children. In both versions, items are rated on a three-point scale from 1 (never) to 3 (often). A lower score represents more optimal EF. The BRIEF-P contains 63 items divided into five scales: inhibitory control, shifting, emotional control, working memory, and planning–organisation. The BRIEF contains more items (a total of 86), representing three additional EF: initiation, organisation of materials, and monitoring. In both questionnaires, the sum of the five or eight clinical scales constitutes the Global Executive Composite (GEC) score. Raw scores are converted to *t*-scores, with higher scores indicating more dysregulation in behaviours associated with EF. For the reliability and validity of both tools, see [20,21].

#### 2.2.3. The POET Intervention

The POET intervention includes eight weekly, face-to-face PT sessions. In the first session, the parents are encouraged to set their own child’s 3–4 functional intervention goals. In each of the following sessions, using the occupational performance coaching (OPC) principles, the parents are coached to identify current and desired performance, barriers, and bridges for achieving their goals [22]. Supporting the daily functioning of children with ADHD requires an understanding of the nature of EF. Therefore, POET adds to the OPC benefits (environmental solutions and occupational adaptations) [22] and specific knowledge and skill acquisition. During this process, parents specifically learn the nature of the identified deficient EF relevant to each occupational goal and strategies to cope with them efficiently. The parents are encouraged to use simple explanations to raise children’s awareness of their delayed EF. While their children perform the intervention goals, the parents assist them to overcome their delayed EF and improve their performance by using cueing, compensations, and/or by teaching their children new skills. For example, the parents learn what impulsiveness and delayed planning are, and why they lead their child to exhibit disorganised play behaviour. Then, they discover constructed movement activities that may be applicable at their home, and how to cue their child to stop a disruptive behaviour and use them when feeling hyperactive. Some parents choose to tell their child: “your body wants to move, let’s try some games that will be enjoyable, and it will also be nice for us to be around you”. All sessions end with prescribing a short and constructed intervention plan for the following week, including up to five strategies. The strategies’ relationships to the children’s EF challenges are well clarified using Barkley’s concepts for EF [4], using professional terms and drawings with symbols, adapted to parents’ learning preferences. If the children’s other developmental delays or parental executive delays are identified as additional barriers that influence the child’s functioning in a specific goal, the written intervention program for each session includes strategies to cope with them, as well [11].

### 2.3. Study Design

The Faculty of Social Welfare and Health Sciences at the University of Haifa (090/13) and Maccabi Healthcare Services, Israel (2013015), provided ethical approval for the study, and the parents signed informed consent forms. Seventeen trained occupational therapists evaluated the participating children and then allocated their parents to a study group (Group A) or a comparison group (Group B) according to their position on the waiting lists (for more details, including a flow of participants through each study stage, see [11]). Because of the ethical commitment to the children’s rights, all participating families received the POET intervention but at different times. Group A started the intervention immediately after the child’s assessment using the CPRS and EFORTS; Group B families waited for 8 to 12 weeks following their first assessment to start the intervention. The waiting period lasted the same as the time needed to apply the POET intervention, which was meant to validate that there were no significant improvements in the study measures without intervention. The occupational therapists directed the parents to set their personal occupational intervention goals for the children using the Canadian Occupational Performance Measure [23]. The first author trained and supervised all the occupational therapists. Following the goal setting, parents participated in 8 to 10 face-to-face PT sessions at the child development centres and occupational therapy clinics from where the children were recruited (for the detailed methodology, see [11]).

One parent of each child (90.3% were mothers) completed the study questionnaires four times at 8- to 12-week intervals between each measure: pre-intervention, post-intervention, first follow-up, and second follow-up for Group A; and pre-wait, pre-intervention, post-intervention, and first follow-up for Group B. Parents were not exposed to the baseline assessment scores of any of the study measure (CPRS, EFORTS) but could not be blinded to the study condition. For a detailed description of the intervention and the POET feasibility testing, including researcher fidelity, see [17].

### 2.4. Data Analysis

We analysed the data with SPSS (ver. 25.0), using descriptive statistics for the demographic information of children and parents. The final sample included 39 families in Group A (study) and 33 in Group B (comparison). We found no significant differences between the two groups’ pre-intervention scores and no improvement in Group B’s scores for any dependent variables (CPRS and EFORTS subscales) after waiting. However, we observed significant differences between Group A Measure 2 (post-intervention) and Group B Measure 2 (post-wait, pre-intervention) for the EFORTS final and subscale scores, *t*(66) = 3.83, *p* < 0.001. Therefore, the two groups were collated to increase the statistical power, and the following analyses pertained to the whole cohort regardless of the original assignment (study or comparison/waiting groups).

Seventy-one families (98.6% of the families who entered the study) completed the intervention but only sixty-five (90.3%) completed all questionnaires at Measure 2 (post-intervention). Fifty-six families (77.8%) reached Measure 3 (8–12-week follow-up), and twenty-nine families (40.3%, all from Group A) reached Measure 4 (26–24-week follow-up). No significant differences were found between parents who completed Measure 4 and those who did not for the children’s gender, age, diagnosis, or pre-intervention CPRS subscale and EFORTS scores. The Shapiro–Wilk test of normality indicated *p* values > 0.05, allowing for parametric statistics.

Then, we examined the first and second hypotheses using multivariate analysis of variance (MANOVA) for repeated measures. Only three measurement points were included due to the low response rate at the second follow-up. Only the CPRS subscales scores directly related to ADHD symptoms (learning, impulsive–hyperactive, and ADHD index) were analysed. No additional ad hoc analyses were necessary.

Unfortunately, we failed to ascertain the proper completion of all the questionnaires administered to the parents. Only 77.78% of the participants completed the EFORTS, and 70.83% completed the CPRS three times. There were no significant differences in the children’s ADHD symptoms (CPRS), age, or study condition between families who completed the study questionnaires at Measure 3 and those who did not. To assess outcomes changes following the intervention, we subtracted the EFORTS and CPRS standardised *t* scores (derived from Measure 1) from the scores obtained in Measure 2. Finally, we applied a stepwise linear regression, including the children’s ages and gender, mothers’ years of education, and the intervention effects over the CPRS subscales as independent variables. This allowed us to examine the explaining variables related to the outcome, as reflected by pre- versus post-intervention differences in the EFORTS. Our prior study identified a significant improvement following the POET intervention in children’s EF as measured by the BRIEF/P scale scores and GEC (Frisch et al., 2019). Therefore, we included the BRIEF/P in the linear regression.

## 3. Results

### 3.1. POET’s Effect on Children’s Ability to Manage Their Daily Routines Efficiently

For the families who completed the EFORTS at three measurements points, significant differences were revealed following the intervention in the EFORTS’s three routine scores, F(6, 200) = 13.87, *p* < 0.001, η^2^p = 0.29, OP = 0.98–1.00, and final score, F(2, 51) = 27.78, *p* < 0.001, η^2^p = 0.52, OP = 1.00, as shown in Table 1.

As depicted in Table 1, the children’s EFORTS scores increased significantly for all three routines following the intervention. No significant differences were found in the EFORTS scores between Measures 2 and 3 (p = ns).

### 3.2. POET’s Effect on Children’s ADHD Symptomatology

We examined the effect on the children’s ADHD symptoms for all the participants who completed the CPRS at the three measurement points (N = 47), using scores for the learning, impulsive–hyperactive, and ADHD index subscales. The MAONVA revealed significant differences between two subscales across the intervention phase, F(12, 176) = 2.71, *p* = 0.002, η^2^p = 0.16, observed power (OP) = 0.99. Table 2 presents descriptive statistics and univariate test values. Both the impulsive–hyperactive and the ADHD index subscale scores decreased significantly following the intervention, and achievements were maintained during the follow-up phase (p = ns).

### 3.3. Variables Contributing to Children’s Improved Daily Routine Management

To elucidate the mechanisms contributing to the gains in daily functioning management, we conducted a stepwise regression analysis for the EF and ADHD symptom score changes following the intervention. The demographic variables (children’s age and gender and mothers’ years of education) examined in the first step did not contribute to the prediction. The variables included in Step 2 were a change in the BRIEF GEC score, an improvement in the impulsivity–hyperactivity scale, and an improvement in the hyperactivity index of the CPRS between Measures 1 and 2. As presented in Table 3, the change in the BRIEF GEC score accounted for 37% of the improvement variance in the EFORTS final score, and the CPRS impulsive–hyperactive subscale score accounted for only 5%.

## 4. Discussion

Previous studies reported the benefits derived from PTs for preschool children with ADHD. These benefits mostly included children’s improved behaviours, parents’ improved behaviours and perceptions, and children’s reduced core ADHD symptoms [12,13]. The current study investigated POET’s efficacy in improving children’s ability to self-manage their daily routine tasks (EFORTS), and in reducing their core ADHD symptoms (CPRS). It also included a secondary analysis to study variables that account for higher percentages of the variation in children’s ability to manage their daily routines following the POET intervention.

Following the intervention, children’s management of daily routines improved, and their ADHD symptomatology was reduced, thus confirming our first hypothesis. Before the POET intervention, the children’s mean scores in all three EFORTS routines and final scores were lower than the EFORTS cut-off scores for the youngest age group [19]. Following the intervention, the children’s EFORTS morning–evening and social routine scores not only showed significant statistical improvement, but they even surpassed published cutoff scores indicating difficulty in this age group. This suggests that the cognitive strategies applied during the POET intervention indeed contributed to a significant clinical improvement in the participating children’s ability to manage their daily routines more efficiently. Earlier studies conducted on interventions for improving EF of children with ADHD measured the positive influence of acquired executive strategies on completing specific occupational intervention goals (e.g., [24,25]). To the best of our knowledge, this study is the first to demonstrate improvements in children’s ability to manage their general daily routine tasks efficiently following a nonmedical intervention. Since the literature shows significant correlations between children’s independence in their routines and parental stress, improving the efficient management of children’s daily routines may reduce adverse childhood experiences of children with ADHD [14]. In this way, POET may contribute to the increased well-being of children and their families and may prevent long-term difficulties across important functional domains [8,26].

Children in the current study also significantly improved their CPRS impulsivity–hyperactivity and ADHD index subscale scores. The learning scale score, measuring attention [27], did not significantly improve following the intervention. One can interpret these findings in the light of the participants’ ages. Most of the current study participants attended kindergarten, and inattentive symptoms tend to become more evident only upon entering structured school settings [28]. However, a few earlier studies measuring the efficacy of PTs on ADHD symptoms did show a significant improvement in attentional symptoms among preschoolers. For example, following the New Forest Parenting Package, a PT intervention specifically developed for preschool children with ADHD, children’s parents rated them as less inattentive and hyperactive [29]. However, an additional study that was focused on improving daily routines in the natural environment among children with ADHD presented results that are similar to ours, though including school-aged children [30]. Mendes and colleagues [30] found direct correlations between children’s independence level in performing daily routine tasks and their hyperactivity severity, but not with the severity of their inattentiveness symptoms. The significant correlation between the concomitant change observed in hyperactivity–impulsivity and daily function management may be attributable to the parents’ priorities. Parents in the current sample prioritised daily dysfunctions highly associated with hyperactivity/impulsivity (e.g., children’s ability to sit for a few minutes during dinner, respond moderately when frustrated, or calmly play a game with a clear purpose indoors [17]). Indeed, some researchers claimed that parents of preschoolers tend to perceive ADHD-derived inattentiveness as less problematic [31] and consider hyperactivity and impulsivity as more impairing [32].

Confirming our second hypothesis, we found that the children’s change in their BRIEF’s GEC scores following the intervention contributed 37% of the variance in the change in their EFORTS scores following the intervention. The CPRS impulsivity–hyperactivity scale explained only an additional 5% of the variance. These findings suggest two interesting insights. First, following an intervention that gave parents tools to cope with their children’s delayed EF in functional contexts, they perceived their children’s impulsiveness and hyperactivity as less interruptive. Second, despite the predominance of impulsiveness and hyperactivity, interventions aimed at allowing children to gain executive control over their daily lives should relate to the children’s wider array of EF, not only their core ADHD symptoms. The order in which the independent variables were included in the regression strengthens this claim. The literature also supports this, recommending that parents of preschool-aged children with ADHD or ADHD-like behaviours consider first turning to PTs and—if they do not progress significantly—combining PT with methylphenidate, aiming at the core ADHD symptoms [2].

A third important insight is suggested. Our previous study demonstrated a significant improvement in children’s EF even before undergoing the BRIEF/P’s clinical cutoff for delayed EF. [1]. All 71 children who participated in this study were included due to their core symptoms and were eventually formally diagnosed with ADHD. In the current study, we demonstrated that what predicted a higher percentage of variance in these children’s enhanced ability to manage their routines was the improvement in their EF, not in their core ADHD symptoms. Therefore, we carefully suggest that this study’s results support early intervention for children with delayed EF, even before crossing the ADHD diagnostic threshold.

Some methodological limitations need to be addressed. A major limitation of the current study is that we had to carry out our study objectives with a sample smaller than the estimated sample. The reason was that 22% of the sample did not complete the questionnaires for Measure 3, and 59.22% for Measure 4. Possibly, this also introduces a Type 1 or 2 error into our results. What additionally decreased the sample size is that by mistake, we did not calculate an assumed dropout rate. All the study measures were based on parents’ reports, which were possibly biased. The meaning is that parents perceived the children as better managing themselves, not necessarily that the objective observation tools would have shown an improvement in children’s EF/ADHD symptoms. For these reasons and due to weak power, we can only carefully suggest that our findings indeed support the recommendation for parent training in preschoolers. Additional limitations: We conducted only expert validity for the 18-item questionnaire that we created based on the DSM-IV text revision to screen the children. We could not randomise the participants to Groups A (study) or B (comparison) because of limitations related to the organisations that recruited the participants. Unfortunately, as reported in a previous study targeting similar objectives [33,34], an attempt to receive teachers’ reports was not successful. Finally, it should be noted that this study, which used a waiting list, should be replicated by using a randomised case control study and extended observations for possible implications for other daily routines and educational setups. We also suggest studying POET’s efficacy in additional cultures, and assessing the results of the intervention when applied by educators in addition to parents.

## 5. Conclusions

This study shows an increase in children’s executive control (self-management) over their daily routines and a decline in their impulsiveness and hyperactivity following the POET intervention. Following our regression analysis results, we suggest focusing on PTs for young children with early ADHD symptoms to cope with these symptoms as well children’s additional executive delays. This may enable children to enhance their daily routine management in their natural environments.

## Figures and Tables

**Table 1 children-10-01083-t001:** Means, standard deviations, F scores, and η^2^p values for the EFORTS questionnaire over time.

Routine	M1	M2	M3	F(2, 94)	Effect Size	
	M (SD)	M (SD)	M (SD)		η^2^p	OP
Morning–evening	2.47 (0.58)	3.11 (0.70)	3.19 (0.72)	47.68 ***	0.48	1.00
Play–leisure	3.13 (0.46)	3.38 (0.62)	3.46 (0.69)	9.45 ***	0.16	0.98
Social	3.03 (0.65)	3.32 (0.60)	3.43 (0.67)	14.83 ***	0.26	0.99
Final score	2.87 (0.47)	3.27 (0.54)	3.38 (0.60)	27.78 ***	0.52	1.00

Note: N = 52. A significant change occurred between Measures 1 and 2 for all routines. Only 52 children had all EFORTS scores for three measures. EFORTS *=* Executive Functions and Occupation Rating Scale; M1 = pre-intervention measure; M2 = post-intervention measure; M3 = 8–12-week follow-up measure; M = mean; SD = standard deviation; η2 = partial Eta squared, OP = observed power. *** *p* < 0.001.

**Table 2 children-10-01083-t002:** Means, standard deviations, F scores, and η^2^p results for CPRS subscales over time.

Scale	M1	M2	M3	F(2, 92)	Effect Size	
	M (SD)	M (SD)	M (SD)		η^2^p	OP
Learning (inattention)	77.34 (15.22)	75.40 (14.41)	73.23 (15.76)	2.60	0.05	0.51
Impulsive (hyperactive)	66.70 11.65	62.72 10.63	63.49 12.20	5.15 **	0.10	0.81
ADHD index	74.89 13.17	69.49 12.33	68.89 12.69	10.47 ***	0.19	0.99

Note: N = 47. A significant change occurred between Measures 1 and 2 for all subscales; no further significant change occurred between Measures 2 and 3 M1 = pre-intervention measure; M2 = post-intervention measure; M3 = 8–12-week follow-up measure; M = mean; SD = standard deviation, η2 = partial Eta squared; OP = observed power. *** *p* < 0.001; ** *p* < 0.01.

**Table 3 children-10-01083-t003:** Stepwise linear regression analysis for post-intervention variables explaining the variance of improvement in the EFORTS final score.

	Model 1			Model 2		
Variable	B	SE B	β	B	SE B	β
Change in GEC post-intervention	−0.03	0.005	−0.61 ***	−0.03	0.005	−5.35 ***
Change in impulsive–hyperactive post-intervention				−0.10	0.006	−0.23 *
R^2^ (Adj R^2^)	0.37 (0.36)			0.05 (0.39)		
F	32.31 ***			4.62 *		

Note: N = 8. EFORTS *=* Executive Functions and Occupation Rating Scale; GEC = global executive composite. * *p* ≤ 0.1; *** *p* ≤ 0.001.

## Data Availability

Data are unavailable due to privacy and ethical restrictions.

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
