# Peer review of "Children with ADHD Symptomatology: Does POET Improve Their Daily Routine Management?"

_children, 2023, doi:10.3390/children10061083_

Round 1

Reviewer 1 Report

I appreciate the opportunity to review this article for Children. I suggest the authors examine the EQUATOR reporting guidelines (according to the study design) to organize the information correctly and not omit necessary details. I provide some data that can be used to improve the manuscript:

Introduction

The introduction of the manuscript is excessively long and the ideas to be presented do not seem to be well spun. In addition, information that should not appear at the beginning of this section, such as the objectives of the study, is shown. I recommend that the authors consult this article when contextualizing their work (doi: 10.1016/j.jclinepi.2013.01.004). The study objectives should be clearly defined.

Methods

Regarding the calculation of the sample size, where are the effect size data taken from? what is the assumed dropout rate?

Also, regarding the participants, if a necessary sample size of 80 participants is defined, why is the study conducted with a lower sample size? where is the selection of the participants made? is the selection by convenience?

Are all the assessment tools adapted and validated for the study population? How many measurement moments are there?

The location of all the information and the order in which it is presented should be reviewed because it is confusing for the reader.

Results

They should be rearranged and reorganized in such a way that they respond directly to the study objective. The analyses to respond to the study objectives are carried out with a sample much smaller than the estimated sample.

Discussion

It is recommended that the authors work on the discussion since it contains aspects to be improved. In the first paragraph, the authors should only show the results of their study. Results that are not mentioned in this first paragraph should not be discussed later.

The results should be interpreted considering "pros" and "cons" taking into account other similar research (if any) and offering a plausible explanation of their results. Findings should be compared with similar studies.

All limitations of the study should be acknowledged, addressing sources of possible bias and inaccuracies....

Conclusion

The study's conclusion should respond to the study hypothesis and, therefore, should be synthesized and more concrete. In addition, it is recommended not to use the verb "to demonstrate", but "to show".

Author Response

I appreciate the opportunity to review this article for Children. I suggest the authors examine the EQUATOR reporting guidelines (according to the study design) to organize the information correctly and not omit necessary details. I provide some data that can be used to improve the manuscript:

Introduction

The introduction of the manuscript is excessively long and the ideas to be presented do not seem to be well spun. In addition, information that should not appear at the beginning of this section, such as the objectives of the study, is shown. I recommend that the authors consult this article when contextualizing their work (doi: 10.1016/j.jclinepi.2013.01.004). The study objectives should be clearly defined

Thank you for the reference. We shortened the introduction and re organized it according to the suggested outlines. We also better defined our objectives.

Methods

Regarding the calculation of the sample size, where are the effect size data taken from? what is the assumed dropout rate?

The sample size was calculated for a research set that includes two groups (study and comparison) and four measurements (Added it on lines 81-82). Unfortunately, an assumed dropout rate was not calculated by mistake, and it is now appearing at the study limitations on lines 379-380.

Also, regarding the participants, if a necessary sample size of 80 participants is defined, why is the study conducted with a lower sample size? where is the selection of the participants made? is the selection by convenience?  

Yes, we relate to it now in line 84. Thank you for your additional comment, we added in the text (lines 102-103): "Due to low recruitment rate, and as the early results were positive, we ended it without retching to 80 participants.

Are all the assessment tools adapted and validated for the study population? Yes, Now added in lines 106-107.

How many measurement moments are there?

Four, as detailed in line 211 (Now highlighted).

The location of all the information and the order in which it is presented should be reviewed because it is confusing for the reader.

We reorganized the information in the Methods section, using the EQUATOR reporting guidelines.

Results

They should be rearranged and reorganized in such a way that they respond directly to the study objective. The analyses to respond to the study objectives are carried out with a sample much smaller than the estimated sample.

 We re-organized the results and omitted results that do not relate directly to the study objectives. As mentioned above we have a gap between the estimated sample size to the sample size used to examine the study objectives. We explained the reason for it on lines 239-240, now highlighted) , and added it as a major limitation on lines 374-376.

Discussion

It is recommended that the authors work on the discussion since it contains aspects to be improved. In the first paragraph, the authors should only show the results of their study. Results that are not mentioned in this first paragraph should not be discussed later.

Done, we include now only results that are discussed later.

The results should be interpreted considering "pros" and "cons" taking into account other similar research (if any) and offering a plausible explanation of their results. Findings should be compared with similar studies.

Done, see on pages 7-9

All limitations of the study should be acknowledged, addressing sources of possible bias and inaccuracies....

Conclusion

The study's conclusion should respond to the study hypothesis and, therefore, should be synthesized and more concrete. In addition, it is recommended not to use the verb "to demonstrate", but "to show".

Corrected on lines 402-405.

Reviewer 2 Report

This is an empirical study of outcome using a psychological intervention called POET to train preschool children in occupational performance or activities of daily living.   The study finds the intervention reduced symptoms of hyperactivity and impulsivity, executive function, and skills as measured by the EFFORTS.   The study went further to try to examine the mechanisms of change and found that change in executive function explained much more of the variance in EFORTS than core ADHD symptoms.   The authors extrapolate from their finding to say that their results support the recommendation for parent training in preschool children prior to use of medication.   The study is well done, makes a clinically meaningful contribution to the field, and is well written. 

I would recommend important revisions to the manuscript.   

1.     The literature review wrongly states that there are no prior studies of organizational skills training.   This is incorrect.  The authors should review the studies of Organizational Skills Training by Richard Gallagher and include this work in their background. 

2.     The authors never describe what skills they are talking about.   What is on the EFORTS?  Sleep?  Self care?   Screens?   Obedience? 

3.     The authors should include in the limitations that 

a.     they developed their own screener for ADHD which has not been validated

b.     the limitations of open label repeated measures, essentially unblinded parent outcome of items that they have directly trained.   This tells you the parent perceived the child as better, not that the child was better. 

c.     A limitation of the finding that hyperactivity was more related to outcomes than IA is no surprise since IA is hard to observe and rate in preschool youth. 

Author Response

This is an empirical study of outcome using a psychological intervention called POET to train preschool children in occupational performance or activities of daily living.   The study finds the intervention reduced symptoms of hyperactivity and impulsivity, executive function, and skills as measured by the EFFORTS.   The study went further to try to examine the mechanisms of change and found that change in executive function explained much more of the variance in EFORTS than core ADHD symptoms.   The authors extrapolate from their finding to say that their results support the recommendation for parent training in preschool children prior to use of medication.   The study is well done, makes a clinically meaningful contribution to the field, and is well written.

Thank you

I would recommend important revisions to the manuscript.   

1.The literature review wrongly states that there are no prior studies of organizational skills training.   This is incorrect.  The authors should review the studies of Organizational Skills Training by Richard Gallagher and include this work in their background.

Thank you for this comment and for the reference. After carefully reading the studied interventions in Gallagher's paper we added on lines 49-53: Few behavioural PTs for children with ADHD symptoms aim at improving the EF of school children. For example, the PATHKO is a performance-based intervention that focuses on ameliorating children's Organization, Time Management, and Planning skills. It was found as promising of clinical utility in improving daily functioning that require OTMP skills among children with ADHD10

To clarify the POET intervention's emphasis we also added on line 62-63: "The POET was developed as an integrated approach that combines performance based practice with skill acquisition in children's natural environment." 

2.The authors never describe what skills they are talking about.   What is on the EFORTS?  Sleep?  Self care?   Screens?   Obedience?

Thank you for this comment! We added an explanation and few examples on lines 142-146.

3.The authors should include in the limitations that 

  a. they developed their own screener for ADHD which has not been validated

  b. the limitations of open label repeated measures, essentially unblinded parent outcome of items that they have directly trained.   This tells you the parent perceived the child as better, not that the child was better. 

  c. A limitation of the finding that hyperactivity was more related to outcomes than IA is no surprise since IA is hard to observe and rate in preschool youth. We cite this fact now on lines- 341-342 

 Thank you, we added those limitations on lines 390-396 (highlighted)

Reviewer 3 Report

The paper is a continuation from a previously published paper (C. Frisch, E. Tirosh, and S. Rosenblum, Phys Occup Ther Pediatr, vol. 40, no. 1, 2020, doi: 473 10.1080/01942638.2019.1640336.) and reports a secondary analysis of data derived from a sample of parents of young children with or at risk of Attention Deficit Hyperactivity Disorder (ADHD) exposed to a parent training intervention delivered by occupational therapists designed to improve the child’s executive function (EF). A far as I can tell the authors started out with the intention of reporting comparative data using a waitlist controlled design. I cannot confirm this because the trial is not registered and the study protocol was not published. Perhaps because of the failure to reach the defined recruitment target the authors have instead reported analyses of pre-post intervention data aggregated from the primary intervention group and the waitlist control group. As such the paper is reporting an uncontrolled open-label trial. A focus was change in ADHD symptoms, and the interaction between ADHD symptoms and executive function. While statistically significant, the change in ADHD symptoms from baseline to post intervention was small (estimated ES 0.19). In contrast the change in EF was medium sized (estimated ES 0.52). A regression analysis identified that change in ADHD symptoms made only a small contribution (7%) to the change in EF symptoms. The authors concluded that interventions directed to EF have an important role in the management of younger children with or at risk of developing ADHD. They also repeat the recommendation made elsewhere that for younger children with ADHD parent training intervention should precede pharmacological intervention. I am sympathetic to the notion of trans-diagnostic approaches to early intervention for younger children with executive function deficits. The conclusions of the authors of this paper need, however, to be tempered given the significant limitations of the study:

1.       Uncontrolled design

2.       Weak power

3.       Use of same informant throughout leading to potential reporter bias

4.       Missing data

Other minor issues:

1.       Abstract should state the type of study ie open label

2.       L27 define ‘young’

3.       L33 Typo. I think you mean ‘extended’

4.       L47-48 There is an essential word missing

5.       L145 Specifiy if power analysis was for pre-post intervention design or for a control trial

6.       L163-166 Note if you had proceeded to an RCT method of randomisation was inadequate

7.       L300-307 Why did you use the hyperactive-impulsive subscale data rather than the ADHD index in your regression analyses?

Satisfactory

Author Response

The paper is a continuation from a previously published paper (C. Frisch, E. Tirosh, and S. Rosenblum, Phys Occup Ther Pediatr, vol. 40, no. 1, 2020, doi: 473 10.1080/01942638.2019.1640336.) and reports a secondary analysis of data derived from a sample of parents of young children with or at risk of Attention Deficit Hyperactivity Disorder (ADHD) exposed to a parent training intervention delivered by occupational therapists designed to improve the child’s executive function (EF). A far as I can tell the authors started out with the intention of reporting comparative data using a waitlist controlled design. I cannot confirm this because the trial is not registered and the study protocol was not published. Perhaps because of the failure to reach the defined recruitment target the authors have instead reported analyses of pre-post intervention data aggregated from the primary intervention group and the waitlist control group. As such the paper is reporting an uncontrolled open-label trial. A focus was change in ADHD symptoms, and the interaction between ADHD symptoms and executive function. While statistically significant, the change in ADHD symptoms from baseline to post intervention was small (estimated ES 0.19). In contrast the change in EF was medium sized (estimated ES 0.52). A regression analysis identified that change in ADHD symptoms made only a small contribution (7%) to the change in EF symptoms. The authors concluded that interventions directed to EF have an important role in the management of younger children with or at risk of developing ADHD. They also repeat the recommendation made elsewhere that for younger children with ADHD parent training intervention should precede pharmacological intervention. I am sympathetic to the notion of trans-diagnostic approaches to early intervention for younger children with executive function deficits. The conclusions of the authors of this paper need, however, to be tempered given the significant limitations of the study:

     1. Uncontrolled design

  1. Weak power
  2. Use of same informant throughout leading to potential reporter bias
  3. Missing data

Thank you for this comment, we have rewritten our conclusions more cautiously and modestly.

Other minor issues:

1. Abstract should state the type of study ie open label

    Added on lines 13

2. L27 define ‘young’

   Added on line 15

3. L33 Typo. I think you mean ‘extended’

Thank you, due to other reviewer's comment we shortened the introduction, and this line was omitted

4. L47-48 There is an essential word missing

 Thank you, added. It is now on line 40.

5. L145 Specifiy if power analysis was for pre-post intervention design or for a control trial

 Added on line 88

6. L163-166 Note if you had proceeded to an RCT method of randomisation was inadequate

We did not proceed to an RCT

7. L300-307 Why did you use the hyperactive-impulsive subscale data rather than the ADHD index in your regression analyses?

We used both, as specified on lines 299-300 (now highlighted)

Round 2

Reviewer 3 Report

Thank you the authors have addressed most of my concerns. I doubt that further revision would add to the quality of the paper.